# *Ai*-Sampler: *A*dversarial Learning of Markov kernels with *i*nvolutive maps

## Abstract

Markov chain Monte Carlo methods have become popular in statistics as versatile techniques to sample from complicated probability distributions. In this work, we propose a method to parameterize and train transition kernels of Markov chains to achieve efficient sampling and good mixing. This training procedure minimizes the total variation distance between the stationary distribution of the chain and the empirical distribution of the data. Our approach leverages involutive Metropolis-Hastings kernels constructed from reversible neural networks that ensure detailed balance by construction. We find that reversibility also implies $C_2$-equivariance of the discriminator function which can be used to restrict its function space.

## 1. Introduction

Markov Chain Monte Carlo (MCMC) is a key approach in statistics and machine learning when it comes to sampling from complex unnormalized distributions. MCMC generates samples by setting up a Markov chain that has the target distribution as its stationary distribution. Once converged, samples can be obtained by recording states from the chain. Not only have MCMC methods transformed Bayesian inference, allowing to sample from the untractable posterior distributions of complex probabilistic models, but they are also widely used for integral estimation, time-series analysis and many other problems in statistics and probabilistic modelling Robert et al. (1999). We argue that this stagnation in developing MCMC algorithms that make use of neural networks is partly due to the well-known difficulty of measuring sample quality (see for example Gorham and Mackey (2015); Brooks et al. (2011)), meaning that systematically establishing the convergence of a chain to its target distribution, or defining a quality measure for samples from a Markov chain is challenging. This makes it intrinsically hard to define an objective function that can be optimized with stochastic gradient descent to improve the performance of the Markov chain. In general, the design of a suitable objective function poses a challenge as one must balance two competing goals: encouraging both *high-quality samples* and *good exploration* of all space Levy et al. (2017).In light of this, we attempt to answer the simple question: *how can we learn to sample from a given unnormalized distribution, using neural network-based MCMC methods?* To answer, we propose a novel MCMC method that makes use of time-reversible neural networks for the transition kernel and derive an upper bound to the total variation distance between the stationary distribution of the resulting Markov chain and the target distribution. The proposed objective makes use of a discriminator. We prove the optimal discriminator to be equivariant with respect to the cyclic group of order 2 and propose a class of $C_2$-equivariant functions that can be used to parametrize it.

## 2. Parametrizing kernels with Involutions

MCMC algorithms are defined by a transition kernel $t(x'|x)$ that maps probability density functions [1] $p_t(x)$ to other probability density functions: $p_{t+1}(x') = \int_{\mathcal{X}} t(x'|x)p_t(x)dx$. The probability density $p(x)$ is *stationary* for the Markov kernel $t(x'|x)$ if

$$\int_{\mathcal{X}} t(x'|x)p(x)dx = p(x'). \tag{1}$$

*Reversible* kernels, namely kernels for which the *detailed balance* condition holds:

$$t(x'|x)p(x) = t(x|x')p(x'), \forall x, x' \in \mathcal{X} \times \mathcal{X}, \tag{2}$$

have $p$ as stationary probability distribution. Throughout the rest of the paper we consider the following transition kernel that satisfies detailed balance with respect to a given probability distribution $p$:

**Definition 1** *(Neklyudov et al., 2020)*
*Given a distribution $p(x), x \in \mathcal{X}$ and a deterministic map $M : \mathcal{X} \to \mathcal{X}$, such that $M \circ M = id_{\mathcal{X}}$, the* involutive Metropolis-Hastings kernel *is*

$$t(x'|x) := \delta(x' - Mx) \min\left(1, \frac{p(Mx)}{p(x)} J_x^M\right) + \delta(x' - x)\left(1 - \min\left(1, \frac{p(Mx)}{p(x)} J_x^M\right)\right), \tag{3}$$

*where $J_x^M$ is the absolute value of the determinant of the Jacobian of $M$ at point $x$.*

As specified in the definition, for this transition kernel to satisfy the *fixed point equation* (1) the deterministic map $M$ must be *involutive*. Given that the kernel is deterministic, this condition would obviously restrict our chain to transition between $x$ and $x' = Mx$. In order to cover the whole support of $p(x)$ we introduce *auxiliary variables* $v \in V$. Then, instead of sampling from $p(x)$ we sample from $p(x, v) = p(x)p(v|x)$, with $p(v|x)$ being any probability distribution we can efficiently sample from. The involutive MCMC framework formulates MCMC algorithms in terms of two degrees of freedom: the involution $M : \mathcal{X} \times V \longrightarrow \mathcal{X} \times V$, and the conditional distribution $p(v|x)$. Since we want to train a sampler that samples optimally, we can define a family of parameterized involutions $M_\theta$ for $\theta \in \Theta$, and we would like to optimize for $\theta$ to obtain a Markov chain that efficiently and effectively samples from a target density. In determining what is efficient and effective sampling, defining what is a natural objective is harder than it sounds. To address this, we design an adversarial game between the mapping $M_\theta$ and a *discriminator*.

## 3. Parameterizing involutions

Recognizing that time-reversibility of deterministic dynamical systems and detail balance of Markov chains are related, we propose as an alternative to use a class of neural networks known as *time-reversible neural networks*.

---

1. Throughout the paper we use the words *density* and *distribution* interchangeably, since we assume every distribution is abs. continuous w.r.t. the Lebesgue measure.

### 3.1. Time-reversible neural networks

In the context of physics-informed learning and forecasting of Hamiltonian systems, Valperga et al. (2022) introduced time-reversible neural networks. These architectures provide good candidates for parameterizing involutions. To motivate this, we first review how (time) reversing symmetries relate to Markov Chains and detail balance, especially in the context of flow maps in Hamiltonian Monte Carlo.

**Reversing symmetries in Hamiltonian MC kernels.** We say that an invertible smooth map $R : \mathcal{X} \times V \to \mathcal{X} \times V$ is a reversing symmetry for an invertible function $L : \mathcal{X} \times V \to \mathcal{X} \times V$ if $R \circ L \circ R = L^{-1}$ We call *time*-reversing symmetry the linear map $R : (x, v) \mapsto (x, -v)$. If $R$ is a reversing symmetry of a function we call the function $R$-*reversible.*Valperga et al. (2022) provide a method for defining parametric functions that, by construction, are reversible with respect to any linear reversing symmetry. In particular, the following theorem holds

**Theorem 2** *(Valperga et al., 2022) Let $L : \mathbb{R}^D \to \mathbb{R}^D$ be an $R$-reversible diffeomorphism[2], with $R$ being a linear involution. Then, there exists a unique diffeomorphism $g : \mathbb{R}^D \to \mathbb{R}^D$, such that $L = R \circ g^{-1} \circ R \circ g$. If $L$ is symplectic, then $g$ can be chosen symplectic.*

This theorem ensures that any $R$-reversible, or in general $R$-reversible and symplectic map $L$, can be decomposed as $L = R \circ g^{-1} \circ R \circ g$. This shifts the problem from that of approximating $L$ to that of approximating the *unique* $g$ of its decomposition. At this point, $g$ can be approximated using any universal approximator with the only constraint that we need the *analytic* form of the inverse $g^{-1}$. Suitable candidates are, for example, compositions of real NVP bijective layers (Dinh et al., 2016), or, as done in Valperga et al. (2022), compositions of Hénon maps (see Appendix D).

**Involutive MCMC kernels by construction.** Using the above result together with definition 1 for the involutive Metropolis-Hastings kernels, we can construct the parametric family of deterministic proposals $M_\theta : \mathcal{X} \times V \to \mathcal{X} \times V$

$$M_\theta = R \circ L_\theta, \text{ with } L_\theta = R \circ g_\theta^{-1} \circ R \circ g_\theta, \tag{4}$$

where $R$ is is the time-reversing symmetry, so that $M_\theta(M_\theta(x, v)) = (x, v)$ for all $(x, v) \in \mathcal{X} \times V$.

## 4. Adversarial training for involutive kernel

Having defined a parameterization for an involutive map $M_\theta$, we are now interested in how to train $M_\theta$ using the transition kernel in equation (3).

### 4.1. Bootstrap

An unbiased estimator of the objective derived in this section can be computed as a Monte Carlo sum over samples drawn *from the target distribution.* Similarly to Song et al. (2017), to get samples from the target distribution and train our transition kernel we propose a bootstrap process that gradually increases the quality of samples over time. We first obtain samples from $p(x)$ using a possibly inefficient and slow-mixing kernel that nonetheless has $p(x)$ as its stationary distribution. The bootstrap process is outlined in Algorithm 1.

---

2. It must be smoothly isotopic to the identity, a mild condition for sufficiently well-behaved target functions.

## 4.2. The adversarsial MH kernel

**Definition 3** *Let $R \circ L_\theta : \mathcal{S} \to \mathcal{S}$ be a deterministic involutive map, and $D : \mathcal{S} \to \mathbb{R}_+$ be a positive valued deterministic function. For a generic test function $r : \mathbb{R}_+ \to [0,1]$, such that $x \cdot r(\frac{1}{x}) = r(x)$, and $r'(x) \geq 0$, we define the* Adversarial Metropolis-Hastings transition kernel *as:*

$$t_D(x'|x) = \delta(x' - R \circ L_\theta(x))r\left[D(x)\right] + \delta(x' - x)(1 - r\left[D(x)\right]). \tag{5}$$

An example of a test function is $r(x) = \min(1, x)$. In this paper, we use the Barker test: $r_B(x) = \left(1 + \frac{1}{x}\right)^{-1}$. In order to sample from the target distribution, we need to satisfy detailed balance, $t_D(x'|x)p(x) = t_D(x|x')p(x')$, with respect to the target distribution. We propose to ensure this by minimising the distance between the target distribution $p(x)$ and the distribution obtained after one step of the chain starting from the target, that is $t_D \circ p(x)$. As done in Neklyudov et al. (2019), we consider the *total variation* (TV) distance:

$$TV[p, t_D \circ p(x)] := \frac{1}{2} \int_{\mathcal{S}} |p(x) - t_D \circ p(x)| dx. \tag{6}$$

For any kernel $t_D$ as in Def. 3 we have $TV\left[p(x'), t_D \circ p(x)\right] = 0$ if $\log D(x) = \log \frac{p(R \circ L_\theta(x))}{p(x)} J_x^{R \circ L_\theta}$. Given that $R \circ L_\theta$ is an involution, we can show that the optimal log-discriminator function has a simple symmetry under the action of $R \circ L_\theta$. We describe next the symmetry and how to include it to derive our final objective.

## 4.3. Equivariance of the discriminator under $R \circ L_\theta$

Let[3] $R \circ L_\theta$ be an involutive map with $R$ volume and density-preserving, i.e., such that $p(RL_\theta(x)) = p(x)$. It's immediate to verify that the density ratio $\lambda(x) = \frac{p(RL_\theta(x))}{p(x)} J_x^{RL_\theta}$ has the following symmetry:

$$\lambda(RL_\theta(x)) = \frac{p(RL_\theta \circ RL_\theta(x))}{p(I \circ RL_\theta(x))} J_{RL_\theta(x)}^{RL_\theta} = -\log \lambda(x). \tag{7}$$

where we used $1 = J_{RL_\theta(x)}^{RL_\theta} J_x^{RL_\theta}$, and $(RL_\theta)^{-1} = RL_\theta$. Therefore, under the action of $RL_\theta$, we would like the parametric discriminators $D$ to transform, *by construction*, similar to the density ratio $\lambda$. We enforce the above constraint using $C_2$−equivariant functions. Let us consider the *cyclic group* $C_2 = \langle g, g^2 = e \rangle$. Note that the action of the –generally non-linear– involution $R \circ L_\theta$ is linear if we consider its action on the lifted space: $x \oplus (R \circ L_\theta(x)) \cong \mathbb{R}^{2n} \oplus \mathbb{R}^{2n}$. In this space, $R \circ L_\theta$ is the representation $\rho_{2n}$ of the $C_2$ group:

$$\rho_{2n} : C_2 \to GL(\mathbb{R}^{2n} \oplus \mathbb{R}^{2n}), \ \rho_{2n}(g) = \begin{bmatrix} 0 & I_{2n} \\ I_{2n} & 0 \end{bmatrix}. \tag{8}$$

To enforce the desired transformation property we parameterize the logarithm of the discriminator with a neural network $d_\phi(x) = \log D_\phi(x)$.

---

3. from now on, where needed for compactness, we omit the composition sign '∘' and simply use juxtaposition.

**Discriminator with product parameterization.** A special construction of $C_2$-equivariant discriminator is with functions $f : \mathbb{R}^{2n} \oplus \mathbb{R}^{2n} \to \mathbb{R}$ that are equivariant under $\rho_{2n}$ and $\xi_1$, namely such that

$$f(\rho_{2n}x) = \xi_1 f(x). \tag{9}$$

Any equivariant function $d : \mathbb{R}^{2n} \oplus \mathbb{R}^{2n} \to \mathbb{R}$ can be decomposed into an equivariant and an invariant part [4]. We can then write the equivariant part as the difference of any function $\eta : \mathbb{R}^{2n} \to \mathbb{R}$ computed at $x$ and $RL_\theta(x)$. The invariant part can be any function $\psi : \mathbb{R}^{2n} \to \mathbb{R}$ of the sum $x + RL_\theta(x)$:

$$
\begin{aligned}
d_\phi(x) &= \psi(x + RL_\theta(x))[\eta(RL_\theta(x)) - \eta(x)], \\
d_\phi(RL_\theta(x)) &= \psi(RL_\theta(x) + x)[\eta(x) - \eta(RL_\theta(x)] = -d_\phi(x).
\end{aligned}
\tag{10}
$$

Note that in this case it is not even necessary to lift the space to achieve the desired equivariance property.

**$C_2$-equivariant composition of linear maps and non-linear activatons.** The more general construction for a $C_2$-equivariant discriminator is with functions $h : \mathbb{R}^{2n} \oplus \mathbb{R}^{2n} \to \mathbb{R} \oplus \mathbb{R}$, which are equivariant under $\rho_{2n}$ and $\rho_1$:

$$h(\rho_{2n}x) = \rho_1 h(x). \tag{11}$$

Let $\hat{d}_\phi$ be a two-channel neural network that we use to approximate the logarithm of the density ratio at both the pre-image and image of $RL_\theta$:

$$
\hat{d}_\phi(x) \approx
\begin{bmatrix}
\log \frac{p(RL_\theta(x))}{p(x)} J_x^{RL_\theta} \\
-\log \frac{p(RL_\theta(x))}{p(x)} J_x^{RL_\theta}
\end{bmatrix}.
\tag{12}
$$

Then, let us consider a linear map $\begin{bmatrix} A & B \\ C & D \end{bmatrix} : \mathbb{R}^{2n} \oplus \mathbb{R}^{2n} \to \mathbb{R}^{2s} \oplus \mathbb{R}^{2s}$, for some $s \le n$. To obtain functions that are equivariant with respect to $\rho_{2n}$ and $\rho_1$ from compositions of such linear maps they must satisfy the following constraint for all $x$:

$$
\begin{bmatrix} A & B \\ C & D \end{bmatrix}
\begin{bmatrix} 0 & I_{2n} \\ I_{2n} & 0 \end{bmatrix}
\begin{bmatrix} RL(x) \\ x \end{bmatrix}
=
\begin{bmatrix} 0 & I_{2s} \\ I_{2s} & 0 \end{bmatrix}
\begin{bmatrix} A & B \\ C & D \end{bmatrix}
\begin{bmatrix} RL(x) \\ x \end{bmatrix}
\tag{13}
$$

which is equivalent to setting $A = D$, $B = C$. We can then compose linear layers of this form with element-wise non-linearities. The function $d_\phi(x)$, that we use to approximate the log-density ratio can then just be the first, or the second, coordinate of the two-dimensional output of $\hat{d}_\phi$.

At this point it is important to notice that in both the proposed parameterizations, the discriminator $D$ depends not only on the parameters $\phi$, but also on $\theta$ through the proposal map $R \circ L_\theta$.

---

4. This is trivial since any scalar equivariant function remains equivariant if multiplied by an invariant function.

## 4.4. Detailed-Balance Loss

As anticipated in the previous sections, we need to minimise $TV[p, t_D \circ p]$. Let $t_D$ be the transition kernel of the type (3) with discriminator $D = \exp(d)$. Given the target density $p(x)$ the total variation distance between $p$ and $t_D \circ p$ is[5]

$$TV[p; t_D \circ p] = |p(RL_\theta(x))r\left[D(RL_\theta(x))\right] - p(x)r\left[D(x)\right]|_1 \tag{14}$$

We consider an upper bound to this quantity.

**Upper Bound on $TV$ with Pinsker Inequality**   We use the famous Pinsker Inequality (Pinsker, 1964) to upper-bound the $TV$ distance with a more malleable KL divergence. Given two densities $p$ and $q$ we have

$$TV[p; q]^2 \leq \mathrm{KL}[p; q]. \tag{15}$$

Considering the final bound, we have the following *adversarial* optimization problem over the parameters $\theta$ of $RL_\theta$ and the parameters $\phi$ of $D_{\phi, RL_\theta}$:

$$\max_\theta A_\theta = \max_\theta \mathbb{E}_{p(x)}\left(r\left[D_{\phi, RL_\theta}(x)\right]\right), \text{with fixed } \phi$$
$$\min_\phi \mathbb{E}_{p(x)}\left(r\left[D_{\phi, RL_\theta}(x)\right]\log r\left[D_{\phi, RL_\theta}(x)\right]\right), \text{with fixed } \theta. \tag{16}$$

Due to the equivariance constraints, $D(x)$ is a function of both $\theta$, through $RL_\theta$, and $\phi$. For any fixed $RL_\theta$ the solution of the minimisation problem is the log-density ratio:

$$D^*(x) = \arg\min \int p(x)r\left[D_{\phi, RL_\theta}(x)\right]\log r\left[D_{\phi, RL_\theta}(x)\right]dx = \log\frac{p(RL_\theta)}{p(x)}J_x^{RL_\theta(x)}. \tag{17}$$

---

**Algorithm 1** Ai-sampler.

---

**Input:** target $p(x)$, initial kernel and discriminator $T_\theta$, $d_\phi$, initial $\mathcal{X} = \{x_j\}_{j=1}^N$, $x_j \sim p_0(x)$, $DiscSteps$, $KernelSteps$.
**repeat**
  $\mathcal{X} = \mathrm{MH}(T_\theta, p, \mathcal{X}, N)$ # MH is the Metropolis-Hastings algorithm.
  **for** X $\in \mathcal{X}$ **do**
    **for** $i = 1$ **to** $KernelSteps$ **do**
      $\theta \to \theta + \epsilon \nabla \mathcal{L}_A(X, \theta, \phi)$ # Eq. (16), first line
    **end for**
    **for** $i = 1$ **to** $DiscSteps$ **do**
      $\phi \to \phi + \epsilon \nabla \mathcal{L}_{adv}(X, \theta, \phi)$ # Eq. (16), second line
    **end for**
  **end for**
**until** convergence
**Return:** $T_\theta$

---

## 5. Experiments

For experiments and pseudo-code description, please see Section A

---

5. We use $|\cdot|_1$ for $\int_\mathcal{S} \cdot dx$

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

## Appendix A. Appendix: Experiments

Following Song et al. (2017), we test our method with the following experiments. The first experiment is with four 2D densities. They highlight specific challenges, such as good mixing in the presence of multiple modes separated by low density regions. The second experiment is with high-dimensional complex densities from real-world scenarios. In particular, we sample from the posterior of a logistic regression model trained with three different datasets of varying number of covariates. For both experiments we benchmark running time and efficiency of sampling of our method.

**Baselines.** For all the experiments we compare with Hamiltonian Monte Carlo (HMC) Duane et al. (1987); Neal et al. (2011) and the method by Song et al. (2017), given its conceptual similarities with ours. For HMC we follow Song et al. (2017) and fix the number of leapfrog integration steps to 40 and tune the step-size to achieve the best performance. A more detailed description of the experiments, including the analytic expression of the densities can be found in the AppendixD and E. Code to reproduce the experiments will be made publicly available upon acceptance.

**Evaluation criteria.** To compare the performances we use the *effective sample size* (ESS). Practically, the ESS is an estimate of the number of samples required to achieve the same level of precision that a set of uncorrelated random samples would achieve (for details see the Appendix F). We report the lowest ESS among all covariates averaged over several trials. Since MCMC methods can be very costly to run, a second often reported performance measure is the *effective sample size per second* (ESS/s). This is simply the ESS obtained per unit of time. We report this measure to evaluate efficiency.

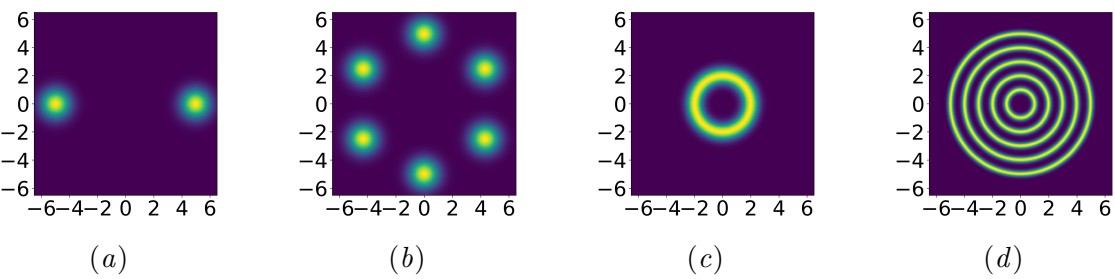

Figure 1: Synthetic 2D densities used in the experiments. From left to right: *mog2*, *mog6*, *ring*, and *ring5*.

### A.1. 2D densities

For a fair comparison with Song et al. (2017) we choose to use the same four 2D densities they used. Two mixtures of Gaussians with two and six modes, *mog2* and *mog6*, a ring-shaped distribution, *ring*, and one made of five concentric rings, *ring5*. The densities are depicted in Fig. 1. For all experiments we ran a single chain for 1000 burn-in steps and compute ESS and ESS/s for the following 1000 steps.

Despite being 2D, these densities pose some challenges. In particular, the mixtures of Gaussians, and the concentric rings are multimodal distributions with high-density regions separated by high-energy (low-density) barriers. This characteristic represents a significant hurdle for Hamiltonian Monte Carlo, as Hamiltonian dynamics are unlikely to overcome these high-energy barriers, potentially leading to inefficient exploration of the state space and convergence issues. Figure 2 shows the very different behaviour of our method compared to HMC highlighting the fast mixing of our method compared to HMC in the presence of high energy barriers. Results are reported in Table 1. See also Appendix F and I.

Table 1: Effective sample size for synthetic 2D energy functions.

| Density | ESS | | |
|---------|-----|-----|-----|
| | HMC | A-NICE-MC | Ai-sampler (ours) |
| mog2 | 0.8 | 355.4 | **1000.0** |
| mog6 | 2.4 | 320.0 | **1000.0** |
| ring | 981.3 | **1000.0** | 378.0 |
| ring5 | 256.6 | 155.57 | **396.5** |

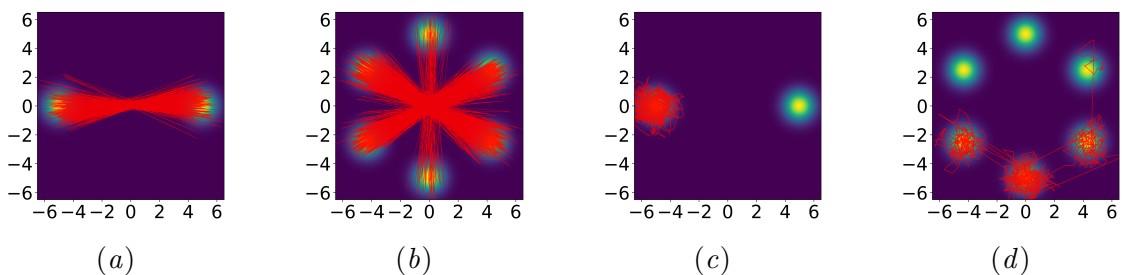

$(a)$ $(b)$ $(c)$ $(d)$

Figure 2: Single MCMC trajectory with the learned kernel (top) on the *mog2* and *mog6* synthetic 2D densities, compared to HMC (bottom). The low density regions make it unlikely for HMC to get from one mode to another.

## A.2. Bayesian logistic regression.

To compare with Song et al. (2017) we use the same posterior distribution they used, resulting from a Bayesian probabilistic logistic regression model on three famous datasets: *heart* (14 covariates, 532 data points), *australian* (15 covariates, 690 data points), and *german* (25 covariates, 1000 data points). For all experiments we ran a single chain for 1000 burn-in steps and compute ESS and ESS/s for the following 5000 steps. Table 2 reports the results.

## A.3. Benchmarking running time

The advantage of learning to sample with a transition kernel parameterized by a neural network also lies in the hardware being optimized to perform multiple operations in parallel. Our method consists of a deterministic proposal parameterized by a neural network and, as

Table 2: Effective sample size for Bayesian logistic regression.

| Density | ESS | | |
|---|---|---|---|
| | HMC | A-NICE-MC | Ai-sampler (ours) |
| heart | **5000.0** | 1251.2 | **5000.0** |
| german | **5000.0** | 926.49 | **5000.0** |
| australian | 1113.4 | 1015.8 | **1746.5** |

opposed to HMC, does not require the gradient of the target density function. In HMC, proposals are obtained by integrating Hamiltonian dynamics which requires multiple calls to the gradient function of the density. For complex distributions, such as Bayesian logistic regression posteriors with large datasets, calls to the gradient are costly.

We implement both our Ai-sampler and HMC in JAX Bradbury et al. (2018) and Flax Heek et al. (2023), making use of the built-in autodifferentiation tools, vectorization, and just-in-time (JIT) compilation. We note that, in a certain range, running time remains approximately constant as the number of parallel chains increases. Past that range, running time increases approximately linearly. We measure running time within the constant range. For both HMC and the Ai-sampler we run Gelman's $\hat{R}$ diagnostic Brooks and Gelman (1998) and find values that suggest good convergence ($\leq 1.004$), and close-to-zero cross-correlation between chains. We can then assume that the ESS of parallel chains is the sum of the ESS of the single chains. For this reason, for a better comparison, we decided to report the ESS per second *per chain*. It is worth noting that, as reported in Fig. 3, we found that the Ai-sampler, given the relatively simple architecture, can sustain many more parallel chains than HMC, which would result in much larger overall ESS. For further details see the Appendix G.

Table 3: Effective sample size per second per chain.

| Density | ESS/s | |
|---|---|---|
| | HMC | Ai-sampler (ours) |
| mog2 | 0.4 | **1052.6** |
| mog6 | 0.98 | **1041.7** |
| ring | **2725.8** | 402.1 |
| ring5 | 333.2 | **434.7** |
| heart | 989.0 | **1736.0** |
| german | 672.0 | **1618.0** |
| australian | 171.95 | **1724.0** |

## Appendix B. Conclusions

We propose the *Ai-sampler*: an MCMC method with involutive Metropolis-Hastings kernels parameterized by time-reversible neural networks to ensure detailed balance. We derive equivariance conditions for the discriminator and a novel simple objective to train the

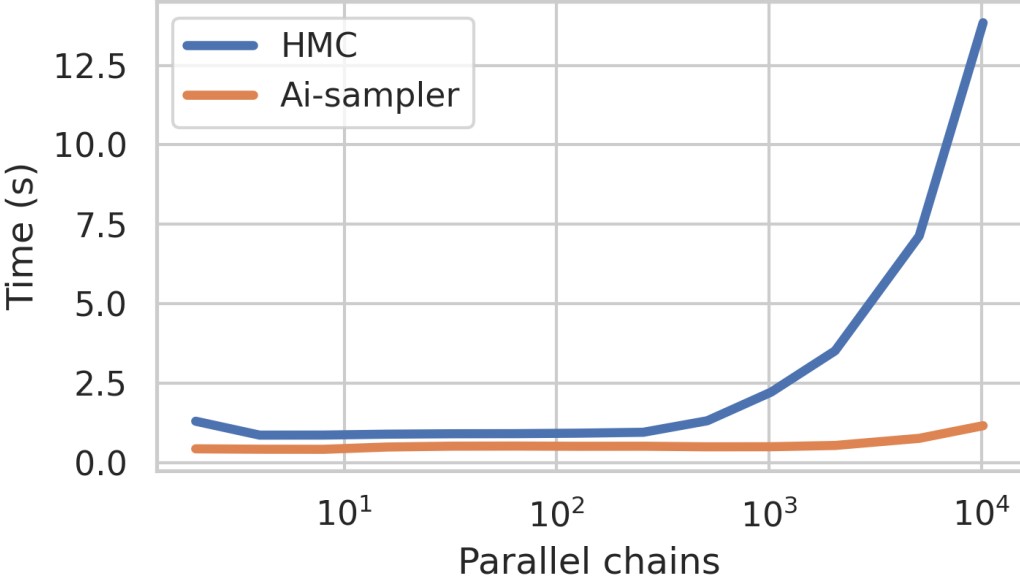

Figure 3: Time vs. number of parallel chains for a single RTX3090 GPU, sampling from the Bayesian logistic regression posterior with German dataset. Every chain consists of 100 steps. For more more than $10^4$ parallel chains, the *jitting* time becomes prohibitively long and therefore not worthy.

parameterized kernel. The proposed objective is an upper-bound on the total variation distance between the target distribution and the stationary distribution of the Markov chain. We use the $C_2$-equivariance of the optimal discriminator to restrict the hypothesisis space of the parametric discriminators. We *learn to sample* with a bootstrap process that alternates between generating samples from the target density and improving the quality of the kernel with the adversarial objective. We demonstrate good mixing properties of the resulting Markov chain on some synthetic distributions and Bayesian inference with real-wold datasets.

In the future, we plan to further explore the potential of our approach as a generative model, given that the adversarial objective is computed from an *empirical distribution*. In this regard, we would like to point out the similarity of our work with Normalizing Flows. In the generative model setting NFs have shown good results and our approach can be applied there two. We expect promising results, as instead of learning a global transformation from a simple probability distribution to a more complex one, e.g., transforming Gaussian noise into good-looking images, our approach consists of learning local transition kernels that map points (e.g., images) to other points in the target distribution.

## Appendix C. Related work

We highlight works that are close to ours in that they make use of neural networks to define MCMC methods. For general MCMC techniques, the reader is referred to comprehensive

surveys such as Roberts and Rosenthal (2004); Brooks et al. (2011); Luengo et al. (2020). To address the challenge in the design of an objective for MCMC methods, previous works Titsias and Dellaportas (2019); Hirt et al. (2021); Roberts and Rosenthal (2009) propose a specific type of deterministic proposal that targets a particular acceptance ratio while promoting mixing with entropy-like regularization. However, this approach imposes restrictions on the type of proposal used and introduces a hyperparameter, that is either the target acceptance rate or the weight of the regularization. Pasarica and Gelman (2010) develop an adaptive MCMC method that selects the parametric kernel that maximizes the expected squared jump. Another solution proposed by Levy et al. (2017) involves optimizing the difference between the average Euclidean distance after one step and its inverse. An interesting method that mixes Variational Inference with Hamiltonian Monte Carlo is the work of Hoffman et al. (2019) where autoregressive flows are used to correct unfavorable geometry of a posterior distribution.A notable method for learning to sample, that is also the closest to our approach, is the work of Song et al. (2017) where a method for training Markov kernels parameterized using neural networks with an autoencoder architecture is proposed. It consists of a GAN-like objective justified by the assumption that GANs attempt to minimize the Jensen-Shannon divergence. Similar to our approach, their method involves a bootstrap process where the quality of the Markov chain kernel increases over time. One key difference is that, to achieve reversibility, Song et al. (2017) make use of an additional random variable, whereas our parameterized deterministic proposals are reversible *by construction*.

## Appendix D. Architectures

All experiments are performed using compositions of parametric Hénon maps. Hénon maps are symplectic transformations on $\mathbb{R}^n \times \mathbb{R}^n$, $(x, y) \mapsto (\bar{x}, \bar{y})$, defined by

$$\begin{cases} \bar{x} = y + \eta \\ \bar{y} = -x + V(y), \end{cases} \tag{18}$$

with $V : \mathbb{R}^n \to \mathbb{R}^n$, and $\eta$ a constant. The reason why we use Hénon maps, other than their approximation properties (see Valperga et al. (2022)), is because they are invertible analytically: for given $V$ and $\eta$ the inverse is simply

$$\begin{cases} x = -\bar{y} + V(\bar{x} - \eta) \\ y = \bar{x} - \eta. \end{cases} \tag{19}$$

For experiments with 2D distributions we use Hénon maps with the function $V$ being a two-layer MLP with hidden dimension 32 and compose 5 of such layers to construct the function $g_\theta$ from Eq. (4). To sample from the Bayesian logistic regression posterior we set the hidden dimension of the two-layer MLP is 64.

For the discriminator we use the product parameterization using two three-layer MLP with hidden dimensions 32 for the experiments with the 2D distribution, and 128 for the Bayesian posterior.

The models have been trained on one NVIDIA A100. Training times are around 2 to 3 minutes for the simple distributions and about 5 to 10 minutes for the Bayesian posterior.

## Appendix E. Analytic form of the densities

Following Song et al. (2017), for all experiments $p(v|x)$ is a Gaussian centered at zero with identity covariance. Now with $f(x|\mu, \sigma)$ denoting the log density of the Gaussian $\mathcal{N}(\mu, \sigma^2)$, the 2D log densities $U(x) = \log p(x)$ used in the experiments are

**mog2:**

$$U(x) = f(x|\mu_1, \sigma_1) + f(x|\mu_2, \sigma_2) - \log 2$$

where $\mu_1 = [5, 0]$, $\mu_2 = [-5, 0]$, $\sigma_1 = \sigma_2 = [0.5, 0.5]$.

**mog6**

$$U(x) = \sum_{i=1}^{6} f(x|\mu_i, \sigma_i) - \log 6$$

where $\mu_i = \begin{bmatrix} 5\sin\left(\frac{i\pi}{3}\right) \\ 5\cos\left(\frac{i\pi}{3}\right) \end{bmatrix}$ and $\sigma_i = [0.5, 0.5]$.

**ring**

$$U(x) = \left( \frac{\sqrt{x_1^2 + x_2^2} - 2}{0.32} \right)^2$$

**ring5**

$$U(x) = \min(u_1, u_2, u_3, u_4, u_5)$$

where $u_i = \left( \sqrt{x_1^2 + x_2^2} - i \right)^2 / 0.04$.

For the Bayesian logistic regression, we define the likelihood and prior as

$$p(\mathbf{y}|\mathbf{X}, \boldsymbol{\beta}) = \prod_{i=1}^{n} \left[ \sigma(\mathbf{x}_i^T \boldsymbol{\beta}) \right]^{y_i} \left[ 1 - \sigma(\mathbf{x}_i^T \boldsymbol{\beta}) \right]^{1-y_i} \tag{20}$$

where $\sigma(z) = \frac{1}{1+e^{-z}}$

Then the unnormalized density of the posterior distribution for a dataset $D = \{\mathbf{X}, \mathbf{y}\}$ is

$$p(\boldsymbol{\beta}|\mathbf{y}, \mathbf{X}, \boldsymbol{\mu}, \boldsymbol{\Sigma}) \propto p(\mathbf{y}|\mathbf{X}, \boldsymbol{\beta}) \cdot p(\boldsymbol{\beta}|\boldsymbol{\mu}, \boldsymbol{\Sigma}) \tag{21}$$

where the Gaussian prior is $p(\boldsymbol{\beta}|\boldsymbol{\mu}, \boldsymbol{\Sigma}) = \mathcal{N}(\boldsymbol{\beta}|\boldsymbol{\mu}, \boldsymbol{\Sigma})$ is a Gaussian with diagonal covariance.

We use three datasets: *german* (25 covariates, 1000 data points), *heart* (14 covariates, 532 data points) and *australian* (15 covariates, 690 data points).

$$p(\mathbf{y}|\mathbf{X}, \boldsymbol{\beta}) = \prod_{i=1}^{n} \left[ \sigma(\mathbf{x}_i^T \boldsymbol{\beta}) \right]^{y_i} \left[ 1 - \sigma(\mathbf{x}_i^T \boldsymbol{\beta}) \right]^{1-y_i} \tag{22}$$

## Appendix F. Effective sample size

Following Song et al. (2017) given a chain $\{x_i\}_{i=1}^N$ we compute the ESS as:

$$\text{ESS}\left(\{x_i\}_{i=1}^N\right) = \frac{N}{1 + 2\sum_{s=1}^{N-1}\left(1 - \frac{s}{N}\right)\rho_s} \tag{23}$$

where $\rho_s$ is the autocorrelation of $x$ at lag $s$. We use the empirical estimate $\hat{\rho}_s$ of $\rho_s$:

$$\hat{\rho}_s = \frac{1}{\hat{\sigma}^2(N-s)}\sum_{n=s+1}^N (x_n - \hat{\mu})(x_{n-s} - \hat{\mu}) \tag{24}$$

where $\hat{\mu}$ and $\hat{\sigma}$ are the empirical mean and variance obtained by an independent sampler.

Following Song et al. (2017), we also truncate the sum over the autocorrelations when the autocorrelation goes below 0.05 to due to noise for large lags $s$.

## Appendix G. Benchmarking time

We report in figure 4 the time it takes to perform one forward pass of the parametric proposal in the Ai-sampler compared to a single call to the gradient function of the Bayesian logistic regression posterior obtained with JAX autodifferentiation. We do not compare

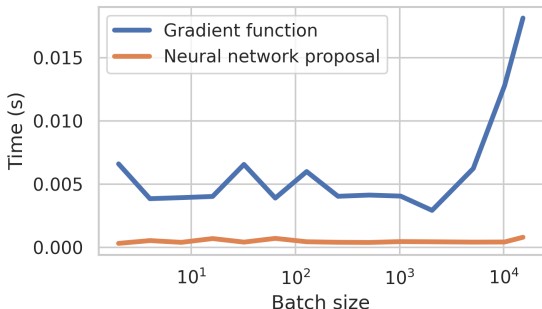

Figure 4: Time taken for a single call of the gradient function and neural network proposal vs. batch size.

running time with Song et al. (2017) as their implementation uses TensorFlow 1 which is not as efficient as JAX, which XLA to compile and run code on accelerators. We stress that time benchmark is to highlight the cost of multiple calls to the density gradient functions, especially in the case of complex Bayesian posterior distributions.

## Appendix H. Additional figures

In Fig. 5 we show an example of training curve, with the acceptance rate, during training.

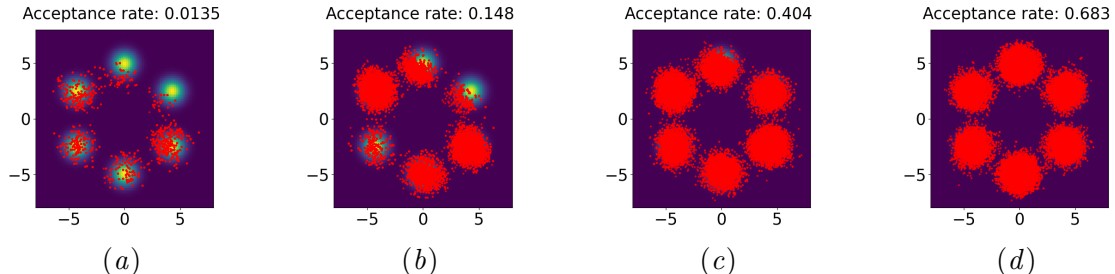

Figure 5: Adversarial objective and acceptance rate during training. Sample quality increasing during training.

## Appendix I. The role of the discriminator

We can investigate how the discriminator is "guiding" the parametric proposal by looking the value of $d_\phi(x)$ after training. In particular, we artificially turn $d$ into a function of two points. For example, for the product parameterization we look at

$$d_\phi(x, y) = \psi(x + y)[\eta(y) - \eta(x)],\qquad(25)$$

for a fixed $x$ and different values of $y$. Figure 6 shows a discriminator, trained with *mog6* for three different values of $x$.

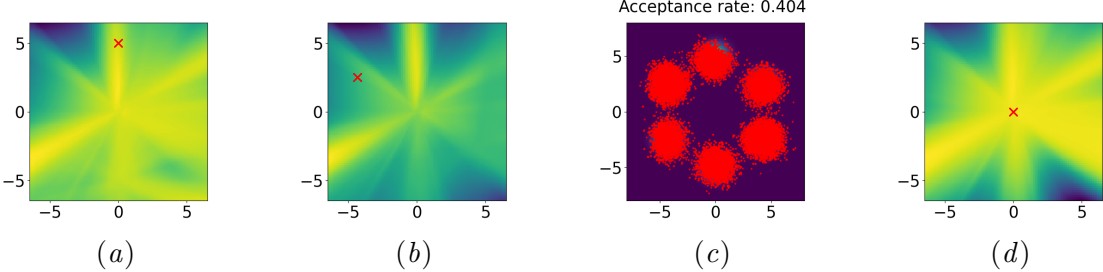

Figure 6: Discriminator artificially turned into a function of two inputs as in Eq. (25), for three different values of x: one far from the six modes and two at the center of one mode.

Note that, for the discriminator to be effective it only needs to be consistent with the ground truth density ratio where the deterministic proposal is likely to propose the new sample from the current state of the chain.

