# OpenReview forum: "Ai-sampler: Adversarial Learning of Markov kernels with involutive maps"
_approximateinference.org/AABI/2024/Symposium — AABI 2024_

### Official Review · Reviewer_ht11 · 2024-04-16
**Interesting revisit of Song, Zhao and Ermon 2017 with a new parametrisation, but problems of structure**

**Rating:** 6
**Confidence:** 3

**Review:**

The manuscript follows the approach of Song, Zhao, and Ermon (2017) to learn efficient proposals for MCMC algorithms via adversarial training and an iterative refinement strategy (called a bootstrap process). The gist of the contribution appears to be an efficient parametrisation of the proposals in terms of involutions, following Neklyudov, Welling, Egorov, and Vetrov (2020), which satisfy detailed balance by construction. Involutions are parametrised using the framework of time-reversible neural networks of Valperga, Webster, Turaev, Klein, Lamb (2022). There is no discussion about ergodicity; the authors should at least include a brief comment.

The contribution is interesting, although only incrementally innovating with respect to the work it builds on. This is not clearly reflected in the "Related work" section, which is a little misleading. Most ideas/tools appear to be already introduced in the three above-cited papers. It is still a relevant contribution to piece them together efficiently.

The numerical results suggest pretty good performance improvements with respect to Hamiltonian Monte Carlo or Song, Zhao, and Ermon (2017).

The presentation of the manuscript is lacking; it really reads like a longer paper which has been compressed and whose end has been put as supplementary information. The authors should try earnestly to reference and comment on the supplementary material in the main text where relevant. In particular, Appendix I looks like it could be interesting, but it is not mentioned anywhere, so explanations are very thin. Authors could possibly move more of the main text to the supplementary material so that the main text is less dense.

- Equation (7) looks like it needs logs on the left-hand side and centre or removal of the log on the right-hand side (with -1 becoming ^{-1}).
- The authors mention the usefulness of considering ESS/s, but they fail to report them consistently. I think ESS are of marginal interest, table 1 and 2 should contain ESS/s
- Why is performance lower than other examples for the ring distribution (table 1)? ESS/s would probably be more meaningful; one could guess that the ring distribution is favourable for HMC, but a comment from the authors would be interesting.
- Are there any convergence issues of the learning task? Mode collapse for the adversarial training? Challenges coming from optimising the upper bound rather than the exact objective function?
- Could you comment on the accuracy of the algorithm? If the loss is not exactly zero, the algorithm might respect detailed balance only approximately and thus not be sampling from the correct target.

---

### Official Review · Reviewer_8n1N · 2024-04-23
**The paper introduces a novel way of parametrizing and optimizing the transition kernel of Markov chains.**

**Rating:** 8
**Confidence:** 2

**Review:**

Strength:
- The paper is theoretically sound, supported by well-justified parameterization choices for the kernel, backed by theoretical justifications.
- The effectiveness of the optimized kernel is demonstrated in simple setups, confirming its effectiveness.

Quality:
- The paper is of high quality and interesting ideas are proposed there.

Originality:
- The paper seems original.

Significance of the work:
- The paper proposes interesting ideas that, on their own, make the work significant in my opinion.
- Although Song et al. tested their method for image generation (a more challenging problem),
- The appendix part of the paper could be extended. Although one can work out the details, the authors could help the reader by providing these details. For example, how Eqs (14, 16) are derived.
- It would be good to comment on how the method performs in image generation tasks. I do not think it needs to be competitive with recent image generation models, but at least it would be good to know where it stands compared to Song et al.

---

### Official Review · Reviewer_bqx8 · 2024-04-24
**Adversarial learning of involutive maps**

**Rating:** 6
**Confidence:** 2

**Review:**

The authors parametrize involutive MH kernels using R-reversible neural networks trained with an adversarial objective. The paper seems to present an interesting and novel direction on MCMC methods and the results seem promising. Given that I am not an expert on MCMC and the topics touched on in this submission, I found the technical details very hard to follow. Moreover, the submission is currently missing a discussion of the experimental results (shown only in the appendix) and the limitations and implications of the work. This might be a very interesting contribution if made more accessible to the broader community.

Minor comments: There are various places in which a space [ ] is missing or some minor typos appeared.

---

### Official Review · Reviewer_d9K6 · 2024-04-24
**Interesting ideas in the space of learnable Markov kernels**

**Rating:** 6
**Confidence:** 3

**Review:**

This work considers learning Markov kernels through *involutive* maps, i.e. a deterministic map $M: \mathcal{X} \to \mathcal{X}$ with the property that
$$
M \circ M = \mathrm{id}
$$
This is done by combining previous works on parameterizable involutive neural networks (Velperga et al., 2022) and on adverserial training of Markov kernels (Song et al., 2017), resulting in a min-max optimization problem involving a *discriminator* $D_{\phi}$ and a learnable map $R L_{\theta}$. The authors demonstrate that this works nicely for toy problems.

Overall, I believe the work is of interest and is well-suited for acceptance to this venue. The text is structured in a slightly strange way, e.g. both "Experiements" and "Conclusion" sections are hidden away in the appendix rather than presented in the main text, and some details are missing, but otherwise the presentation is good. I would be willing to bump my score to a 7 if the structure of the paper can be improved.

Some immediate questions / feedback:
- In the abstract it's mentioned that TV distance is minmized, but from the main text it seems an upper-bound is used; specifically, the KL divergence. I therefore find the claim in the abstract to be somewhat disingenuous and would prefer it if the authors clarified this in the abstract.
- Why use the Barker test rather than Metropolis, which is known to be more effective? I assume it's because Barker's allows gradient computation through the conditional expectation. If so, then this should be explicitly mentioned so the reader is aware that the proposed method won't "just work" with all standard choices of $r$.
- I would very much suggested moving the empirical section (at least a shortened version) and the conclusion to the main text rather than putting it in the appendix. Much of the more theoretical aspects can easily be moved to the appendix instead.
- Figure 2: there's no "top" and "bottom" so the caption is confusing. I'd also recommend showing this figure in the main text.
- Does the ESS / s metrics include the training time? If not, this should be explicitly caveat-ed.
- I would mention in the main text somewhere what choice of $p(v \mid x)$ is made in the experiments, e.g. "In this paper, we follow Song et al. (2017) and use a zero-centered unit-variance Gaussian for $p(v \mid x)$." or something. It's an immediate question a reader will have, so I believe it's useful to make this explicit immediately.

---

### Official Review · Reviewer_ruxR · 2024-04-25
**Very interesting idea, deserves attention and further development.**

**Rating:** 8
**Confidence:** 4

**Review:**

This paper is about designing reversible Metropolis-Hastings transition kernels for MCMC that yield much faster mixing using neural nets. One key insight is that they define a deterministic transition kernel such that the two applications of the kernel transition always comes back to the same point, hence the term involutive map.  Moreover, the authors note further equivariant structure of the class of involute maps, which can further inform and restrict the space of maps one has to search over. The challenging part seems to be parametrizing the neural network associated with the transition map, and the adversarial training. My main criticism is that the explanation of the training objective is rather opaque -- I think the paper would benefit from more discussion about the training procedure. I think details about the involutive map for the adversarial kernel can be left to the appendix.

Examples are interesting, this idea deserves further attention and development. I am curious as to why the rings example performs not very well for the Ai-sampler. Perhaps it's due to the continuous symmetry in the ring? I am also curious about the stability of the method especially with regard to the adversarial training aspect. My impression is that training maps in an adversarial way is often unstable unless the neural net has addition structure (see GAN vs WGAN for example). I appreciate the authors trying the method on a high dimensional example, but Bayesian Logistic regression yields posteriors that resemble Gaussians, so I do not gain much insight into how well the method scales in high dimension.  I think the method would shine most for complex multimodal distributions.

---

### Meta-Review · Area_Chair_zkVy · 2024-05-13

**Recommendation:** Accept (Poster)
**Confidence:** 5

**Metareview:**

The paper proposes a new method to learn the transition kernels by neural networks in MCMC. Reviewers found the kernel construction methods interesting and promising.

---

### Decision · Program_Chairs · 2024-05-27

Accept